# Neuroergonomic Attention Assessment in Safety-Critical Tasks: EEG Indices and Subjective Metrics Validation in a Novel Task-Embedded Reaction Time Paradigm

**DOI:** 10.3390/brainsci14101009

**Published:** 2024-10-07

**Authors:** Bojana Bjegojević, Miloš Pušica, Gabriele Gianini, Ivan Gligorijević, Sam Cromie, Maria Chiara Leva

**Affiliations:** 1Human Factors in Safety and Sustainability (HFISS), Technological University Dublin, D07 EWV4 Dublin, Ireland; bojana.bjegojevic@tudublin.ie (B.B.);; 2Centre for Innovative Human Systems (CIHS), Trinity College Dublin, D02 PN40 Dublin, Ireland; 3mBrainTrain LLC, 11000 Belgrade, Serbia; ivan@mbraintrain.com; 4Department of Informatics Systems and Communication (DISCo), Università degli Studi di Milano-Bicocca, 20126 Milan, Italy

**Keywords:** attention, neuroergonomics, EEG, engagement index, beta/alpha ratio, reaction time, coefficient of variation, attention ratings, MATB-II, safety-critical tasks

## Abstract

**Background/Objectives:** This study addresses the gap in methodological guidelines for neuroergonomic attention assessment in safety-critical tasks, focusing on validating EEG indices, including the engagement index (EI) and beta/alpha ratio, alongside subjective ratings. **Methods:** A novel task-embedded reaction time paradigm was developed to evaluate the sensitivity of these metrics to dynamic attentional demands in a more naturalistic multitasking context. By manipulating attention levels through varying secondary tasks in the NASA MATB-II task while maintaining a consistent primary reaction-time task, this study successfully demonstrated the effectiveness of the paradigm. **Results:** Results indicate that both the beta/alpha ratio and EI are sensitive to changes in attentional demands, with beta/alpha being more responsive to dynamic variations in attention, and EI reflecting more the overall effort required to sustain performance, especially in conditions where maintaining attention is challenging. **Conclusions:** The potential for predicting the attention lapses through integration of performance metrics, EEG measures, and subjective assessments was demonstrated, providing a more nuanced understanding of dynamic fluctuations of attention in multitasking scenarios, mimicking those in real-world safety-critical tasks. These findings provide a foundation for advancing methods to monitor attention fluctuations accurately and mitigate risks in critical scenarios, such as train-driving or automated vehicle operation, where maintaining a high attention level is crucial.

## 1. Introduction

Attention is “[…] a state [of] optimal level of activation that allows selecting the information we want to prioritize in order to control the course of our actions [1]” (p. 184). It is known to fluctuate constantly [2,3], which poses challenges to maintaining optimal performance. Suboptimal levels of attention are associated with many negative outcomes, particularly in safety-critical scenarios. For instance, it is considered the main cause of car accidents [4]. With the increase in automation in human–machine systems, sustaining attention to the task is becoming increasingly challenging [5]. To address this issue, Industry 5.0 is focused on implementing adaptive automation solutions, ensuring that systems can respond dynamically to operators’ fluctuating levels of engagement and attention. Feeding the adaptive automation solutions with the real-time information regarding operators’ attention level can mitigate issues of fatigue and excessive or suboptimal workload, leading to the improvement of safety and effectiveness in human–machine systems [6,7]. Therefore, defining accurate and valid indicators that reliably reflect these fluctuations in attention level in real time is an essential task in neuroergonomics, a science of the human brain in relation to performance at work and everyday settings [8,9,10,11].

While the *object* of the current focus of attention can be readily detected using wearable eye-tracking technologies, detecting the *level* of attention allocated to the task poses greater challenges. Continuous innovations in the design of wearable electroencephalograms (EEGs) and brain–computer interface (BCI) technology offer a promising solution by providing continuous measurements of the fast-paced electrical activity of the brain on a millisecond timescale. Decreasingly cumbersome designs such as self-adhesive EEG tattoos [12], EEG-integrated headphones [13,14], elastic headbands [15], and in-ear [16] and cap designs [17] are overcoming the problem of obtrusiveness and enabling seamless integration with normal daily activities. These designs also eliminate the need for expert setup. In addition, the developments in computational intelligence and algorithms for the online pre-processing and de-noising of the EEG signal, such as artifact subspace reconstruction (ASR) [18,19], are enabling its use in real-world and dynamical environments, even while moving.

Despite the technological advancements equipping us with the tools for acquiring brain signals in real-world environments, the methodological guidelines on the use of EEG for tracking attention level “in the wild” are lacking. One of the most commonly used methods for evaluating mental states in dynamic tasks and real-world scenarios are frequency-based metrics obtained by transforming the continuously recorded EEG signals (during specific task execution) from time-domain into frequency-domain. Typically, these signals are described in terms of frequency bands such as delta (0.2–3.5 Hz), theta (4–7.5 Hz), alpha (8–13), beta (14–30 Hz), and gamma (30–90 Hz) [20]. These bands were somewhat arbitrarily defined, but factor analysis of EEG spectrograms showed a substantial overlap with classically accepted frequency ranges [20]. In most studies, theta, alpha, and beta frequencies have been examined in relation to attention and associated cognitive processes [21,22]. In the equations below, they are denoted as δ, θ, α, and β, respectively.

EEG frequency bands are often combined into a ratio, indicating the relative power of specific bandwidths associated with particular mental states, and referred to as an index. The engagement index is one such metric considered to reflect mental engagement [5,23], attention, vigilance, and/or alertness [24,25]. While different metrics can be found in the literature under the name of engagement index, referring to different means of computation, this study focuses on two variants most frequently used. The first variant, originally proposed by Pope et al. [5], is computed as the ratio between slower brain waves in the alpha and/or theta ranges (associated with lower alertness and attention) and faster ones in the beta range (associated with cognitive processing and higher mental engagement). Since the initial studies [5,26] compared different versions of this index and concluded that β/(α + θ) best reflected user’s engagement, we refer to this variant as the engagement index (EI). The second version, described by the same authors [5,26], omits the theta power from the equation, representing the ratio of beta to alpha power and is referred to as beta/alpha (or Β/A for short in the present study).

Both of these indices have been used in a variety of contexts, ranging from lab-based to real-world tasks scenarios [23,25,27,28,29,30,31,32,33,34]. However, while the application of EEG-based indices in assessing attention and engagement levels in dynamic environments is a growing field of interest, the methodological rigor and empirical validation required for their reliable use remain underdeveloped. Although the engagement index (EI) has been used in various contexts as an indicator of focused attention and mental engagement, its sensitivity to dynamic changes in attentional demands and its effectiveness in predicting attention lapses in safety-critical environments remains under-explored.

### 1.1. Related Work

EEG indices, including the engagement index (EI) and beta/alpha ratio (Β/A), have been applied in various contexts and have been associated with somewhat related yet distinct cognitive concepts. For instance, the EI, accompanied by certain proprietary algorithms, is used in NASA’s commercial neurofeedback device *Narbis glasses*, designed to aid in controlling and increasing the wearer’s focus to the task (https://www.narbis.com/, accessed on 29 July 2024). Similarly, it has been used for detecting drops in engagement and vigilance in lab tasks, such as the Conners’ Continuous Performance Test (CPT) [29], as well as in educational contexts, where it was integrated as part of a real-time biofeedback device [25]. This device constituted an EEG headset and a scarf that was supposed to deliver haptic feedback whenever a drop in engagement was detected. However, apart from focused and sustained attention, EI has been used in mental workload detection [35] and was found to be the best feature for neural network classification of workload during multitasking in NASA’s Multi-Attribute Task Battery (MATB-II) [36]. Additionally, it has been demonstrated to differentiate between two levels of engagement efficiently in real-flight conditions, with increased EI in difficult flying conditions leading to a higher miss rate in detecting auditory probes [27].

While EI has been applied across various contexts, the beta/alpha (Β/A) ratio is used in an equally if not even more diverse fields of application; namely, in addition to lab-based attention tasks such as CPT [23,29], Β/A was used in cognitive tasks and virtual reality (VR) as a stress indicator. In particular, in the study by [34], Β/A values of 1.5 or above were considered to be indicative of stress during tasks such as reading, writing, and problem solving. Similarly, another study explored the effects of a VR roller coaster ride as a method of stress induction on the inverse of Β/A, i.e., the alpha/beta ratio, revealing negative correlations with stress level [33]. Furthermore, Β/A has been used in clinical contexts for diagnostics of cognitive decline and dementia [32], as well as sustained attention performance in patients with traumatic brain injury [29]. Moreover, in neuromarketing research it has been found to be a useful indicator of banner preference and efficiency [31].

Few studies have explicitly addressed the question of the validity of EI and Β/A as indicators of attention level. For instance, in the study in [25], where EI was used to detect the level of engagement in real-time, it was calibrated for each participant by considering the lowest values during relaxation periods and the highest during arithmetic problem-solving. While this approach has merit in determining the individual ranges of EI for real-time use in specific contexts, its predictive validity and reliability were not examined within the same experimental conditions. Kamzanova et al. [28] conducted a study that addressed the adequacy of EI and other EEG indices for monitoring attention during a 40-minute vigilance task for the purpose of early detection of vigilance loss. Importantly, they tested which index was most diagnostic of vigilance decrement under two levels of workload and found that lower-frequency alpha bands were more sensitive to changes in task parameters than the EI. However, the authors recommended further research to explore how the diagnosticity of EEG indices varies with task demands.

There is somewhat stronger evidence supporting the association between the beta/alpha ratio and attention levels. Specifically, Β/A is considered as an index of degree of concentration by [32] as it seemed to differentiate between healthy subjects and those with mild cognitive impairment and Alzheimer’s dementia successfully, serving as a promising tool for the early detection of dementia. Additional support for Β/A as an indicator of attention level comes from a lab-based sustained attention study [23], which demonstrated negative correlations between Β/A and both mean reaction time and its variation in the CPT task, which are the most reliable objective proxies of attention [3,37,38].

However, these studies mainly involved static lab tasks that do not reflect the dynamic nature of real-world environments. The lack of consistent validation across various contexts raises concerns about the generalizability and reliability of these indices in more complex, dynamic scenarios, which is crucial for their application in safety-critical environments. As underscored by [39], EEG indices should not be used blindly but must always be empirically validated with respect to their sensitivity, especially in automated systems designed to monitor the operator’s state. Furthermore, the lack of experimental manipulation of attention levels prevents stronger conclusions regarding the relationship between these indices and the level of attention.

In conclusion, while the EI and Β/A ratio have shown promising results with respect to reflecting attention levels in controlled environments, their applicability in dynamic, safety-critical tasks remains underexplored and not fully established. This study aims to fill this gap by testing the sensitivity of these indices in a novel, task-embedded reaction time paradigm that allows for the validation of different measures by gradually manipulating the level of attention and continuously sampling the reaction times as an objective indicator. This approach more closely reflects real-world dynamic conditions while maintaining experimental control. By doing so, it seeks to provide a more nuanced understanding of attention and engagement in safety-critical contexts, contributing to the development of more reliable methods for dynamic attention assessment in real-world tasks.

### 1.2. Research Overview

This study addresses the dual goals of (1) developing and validating a dynamic, naturalistic paradigm for assessing attention and (2) evaluating the effectiveness of EEG indices and subjective measures in reflecting attention levels in safety-critical scenarios. To achieve these goals, we followed the methodological steps outlined in Figure 1. The paper outline follows the same structure and is described below.
We adapted the NASA Multi-Attribute Task Battery (MATB-II), a commonly used computer-based task in neuroergonomics studies, to manipulate the level of attention required gradually to cope with the task successfully. This was achieved by setting a simple reaction time task as a primary task that participants had to engage with throughout the experiment, which provided a baseline level of attention to this task without any multitasking. Two secondary tasks were added incrementally to increase the required level of attention in order maintain task performance.By gradually adding the secondary tasks (each combined with the primary task only, and all three tasks combined), four distinct levels of attentional demands were created (low; medium 1; medium 2; high). To examine the impact of sequence of attentional demands on attention, the sample was divided into two groups, presented with either increasing or decreasing demands, while controlling for fatigue.We recorded the physiological signals (EEG, eye-tracking, and ECG; to maintain the text at a comprehensible length, only the EEG signals are discussed in the present paper), performance, and subjective measures of 60 participants while engaging with the task. The EEG data were pre-processed, and two versions of engagement index were extracted: β/(α + θ) and β/α.The experimental paradigm, including the experimental manipulation and fatigue control, was tested and confirmed, establishing the groundwork for EEG indices evaluation.The EEG indices were statistically evaluated in their response to different attentional demands as well as to the dynamics of changes in those demands operationalized as increasing vs. decreasing demands.The relationship between EEG indices and performance-based measures was examined by testing their overall correlations, as well as within each experimental condition, to gain more nuanced insights into the patterns of their sensitivity.Finally, several regression models, combining all the measures (EEG, performance, and subjective ratings), were run to explore the differential contribution of each measure to predicting the attentional lapses. To this end, we used both a modified Poisson regression and a number of library models whose hyperparameters were chosen using an AutoML procedure. Furthermore, we assessed the input feature importance using their Shapley value.

Our hypotheses were as follows:

**H1.** 
*Engagement indices will increase with higher attentional demands (as reflected by experimental conditions).*


**H2.** 
*Engagement indices will correlate with reaction times (RTs) or coefficient of variation (CV) of RTs across different task conditions.*


**H3.** 
*EEG-based engagement indices will provide more predictive information regarding attention compared to subjective measures.*


Our findings indicate the differential sensitivity of the two engagement index variants to various aspects of attention. The β/α ratio appeared more sensitive to changes in attentional demand, while the EI, including θ, may reflect the effort to sustain attention and performance. With respect to the regression analysis on the omissions (representing attention lapses), the resulting Pearson correlation coefficient between the true and the predicted variable from the best model was close to 0.96. In terms of the Shapley value of the input features, we found that the most impactful features were the average reaction time and its coefficient of variation.

To the best of our knowledge, this study is the first to evaluate and compare these two common variants of the engagement index in response to varying attentional demands and assess their validity by comparing them with objective and subjective attention indicators, bridging the gap between controlled experimental settings and real-world applications. Our results suggest that combining these two indices along with behavioral and subjective measures, as well as contextual information, can provide a more nuanced understanding of attention levels in safety-critical scenarios.

## 2. Materials and Methods

### 2.1. Participants

The study involved 60 participants (mean age = 30.49 years; SD = 7.53; 22 females), mostly postgraduate students. Participation was on a voluntary basis and non-incentivized. All participants signed an informed consent form prior to the start of the experiment. The study was approved by The Research Ethics and Integrity Committee of the Technological University Dublin under the reference document REIC-21-48.

### 2.2. Experimental Paradigm: Task-Embedded Reaction Time

To validate a cognitive state measure, the following components are usually required, according to [40,41]:(a)Experimental manipulation of task demands to evoke the change in cognitive state of interest;(b)Objective measurements of performance metrics (reaction time, errors etc.);(c)Subjective measures capturing participants’ perception of level of difficulty, effort, etc.

To achieve the first requirement, the NASA Multi-Attribute Task Battery (MATB-II) was selected. As described in detail in Section 2.2.1., this task was specifically adapted to manipulate the level of attention needed experimentally to perform the task successfully. To obtain the objective performance indicator, reaction time and omissions were measured. Embedding a reaction-time task into a multitasking scenario allows for continuous, real-time measurement of attention fluctuations. Finally, along with the EEG and performance metrics, subjective measures were recorded, as detailed in Section 2.4.

Importantly, to ensure that the observed effects of the task are indeed a consequence of different attentional demands, it is also essential to control for important confounds such as fatigue, mental workload, and attentional load sequence. With respect to the control of fatigue levels, the task duration was kept under 30 min. Time spent on the task, in addition to the task demands, is known to be a contributing factor to performance fluctuation and decline, including the increase in reaction time and decrease in accuracy, due to mental fatigue [42,43]. Furthermore, the baseline condition was repeated at the end to measure if any fatigue effects occurred despite shorter task duration.

Mental workload is inextricably intertwined with engagement and attention in naturalistic tasks, as well as real-world scenarios (one such example is demonstrated in the study by [27]), making it difficult to disentangle their effects. However, attention levels can vary irrespective of workload levels, such as in sustained attention tasks and monotonous conditions. Since these two concepts reflect different aspects of cognitive functioning, efforts should be made to differentiate between carefully them as much as possible. According to [42], multitasking is one of the means through which mental workload or cognitive load can be modulated. In our task, this modulation of the load is mediated by attention, since switching between multiple tasks requires attention switching. As explained in Section 2.2.2., describing the task adaptation, none of the subtasks is overly cognitively demanding. It is the requirement to multitask, i.e., to switch attention between them, that is demanding. This poses a load on the attentional resources, or, in other words, creates different levels of attentional demands. Regardless, to ensure that our paradigm is indeed valid for capturing the changes in attention level, we have extracted the EEG-based mental workload index (MWI). This metric is considered a gold standard in neuroergonomics and has been shown in numerous studies as a reliable indicator of mental workload [6,10,13,44]. If the MWI demonstrates greater sensitivity to the changes in experimental conditions, this could imply that our task is predominantly sensitive to variations in cognitive load, rather than the level of attention. Therefore, our hypotheses were as follows: (1) that MWI will be less sensitive to the task conditions compared to the engagement index; (2) that MWI will not correlate significantly with reaction time to the primary task.

Finally, different dynamics or sequences of task demands were shown by previous studies to impact the physiological, behavioral, and subjective indicators of cognitive constructs, such as attention and workload, significantly [45]. Therefore, the sequence of task demands (increasing or decreasing) was incorporated as an independent variable into the study design.

#### 2.2.1. Procedure

The experiment was conducted on an individual basis, in a quiet room, with no source of natural light and consistent luminance levels. Upon signing the informed consent, the participant was presented with a video containing detailed task instructions, followed by a short quiz and a practice session, to ensure that the task requirements were fully understood (Figure 2). This was followed by equipment setup and calibration. Before the start of the MATB task, as well as at the end of it, participants are asked to provide a self-assessment of their focus and tiredness levels, which they were previously informed of. This is repeated throughout the task, once per each condition, along with the subjective task difficulty assessment. The order of the questions was randomized to avoid automatised responses. Depending on the group they were assigned to, they were presented with either an increasing (G1) or decreasing (G2) task demands, which they were not aware of.

#### 2.2.2. MATB-II Task Adaptation

For the purpose of manipulating the attention level required to cope with the task, we adapted the NASA MATB-II task, originally devised for human performance and workload research [46]. The details of task adaptation are also presented in [47]. The main objective of this adaptation was to vary the level of task demand gradually as this is expected to evoke a corresponding degree of mental engagement and attention [5].

In essence, three subtasks were used and combined in such a way to create a gradient of attention demands (3). The task paradigm consisted of system monitoring (SysMon; top left rectangle in Figure 3A), set as a primary subtask to which other subtasks were added as secondary tasks. The goal of SysMon is to maintain the constantly oscillating scales in the center of each of the four bars. The MATB-II includes a training mode which simplifies this subtask by showing a small indicator next to the scale that needs adjusting, which, in turn, is achieved by simply clicking on the scale in question with a computer mouse. By keeping the training mode throughout the task, the decision-making aspect was eliminated to allow for simple reaction time measurement. The appearance of indicator pointing to a particular scale is denoted here as an “event”, was randomized both temporally (2–8 s) and spatially (across the four bars), and had a visually displayed lapse time of 10 s. There were 60 events presented in each condition, totaling 300 reaction time measurements. In the instances when the participant did not click on the required scale within the 10 s, this was marked as an omission.

Irrespective of the sequence of multitasking conditions (group 1: increasing demand; group 2: decreasing demand), both groups of participants were initially presented with condition A1, which solely comprised the primary (SysMon), to obtain the baseline reaction time of participants. The same condition was repeated at the end (labeled as A2, Figure 3B) to allow for measuring the effects of fatigue. The multitasking conditions B, C, and D were developed by adding either the communications subtask (Comm), resource management (ResMan), or both, respectively (Figure 3). The Comm subtask involves listening to audio messages and, in the instances when the aircraft label is called out, tuning the required radio to the specified frequency. The participant need not engage with this task until an audio message starts, which makes this subtask a bottom-up type of distraction from the primary task. In addition, 5 out of 15 Comm messages were “false alarms”, which did not refer to the aircraft label and therefore did not require any action from the participant’s side. On the other hand, the ResMan subtask requires constant monitoring and engagement to maintain the fuel levels in the two main tanks at a required level by clicking on different pumps and transferring the fuel. To make this task more demanding, the pumps between the infinite-source tanks and the main tanks were disabled. Each condition lasted about 5.5 min, therefore keeping the entire task duration under 30 min to avoid confounding effects of fatigue on attention decrements.

### 2.3. EEG Data Acquisition and Processing

For recording the brain activity during the task, a semi-dry wireless EEG (mBrainTrain) with 24 channels and a sampling frequency of 500 Hz was used. The Ag/AgCl electrodes are embedded in a cap adhering to the international 10–20 system of electrode placement [48]. The cap utilizes sponges immersed in saline solution for conducting the brain signals. The electrodes were referenced online to the FCz electrode, and AFz served as the ground electrode. Online filters were applied for visualization purposes only. The electrode impedance was set to below 10 kΩ during the preparation and setup.

The raw signals were bandpass filtered offline between 1 and 40 Hz, followed by the removal and interpolation of flat and/or poor signal quality channels (the mean number of such channels per participant was <1). Artifact removal was achieved via artifact subspace reconstruction (ASR), setting the ASR parameter to the value 13 [18,19]. Since the chosen parameter was not overly strict, residual blink artifacts, lateral eye movement artifacts, and EMG, ECG, and specific-channel noise were eliminated using the infomax independent component analysis [49] (ICA) algorithm (in the implementation provided by the EEGLAB pop_runica() function [50]). Prior to running the ICA, re-referencing the average reference was conducted, as recommended in [51]. Since the number of interpolated channels, as well as the average re-referencing, cause an inevitable loss of unique information due to the linear dependency between certain data points, the number of computed independent components (ICs) was adjusted to align with the resulting reduction in the data rank [52]. The mean number of ICs removed per participant was 2.5, ranging from 1 to 7. In the analysis of EEG data, a sliding window of 2 s was used, with a 1 s step size.

### 2.4. Measures

Following the neuroergonomics triangulation approach for measuring a specific cognitive function/state, we have collected subjective, behavioral, and physiological measures. Specific metrics are listed in the respective sections below.

#### 2.4.1. Subjective Measures

Three subjective ratings were recorded during the experiment: (1) subjective level of focus (attention); (2) subjective level of tiredness (fatigue); and (3) subjective level of task difficulty. All three were recorded online, in response to an auditory prompt, similar to [40,45,53], and on a Likert scale from 1 to 10, avoiding a central indifference point [54]. The ratings were recorded once halfway through each condition, as well as before and after the start of the task for the subjective focus and fatigue.

#### 2.4.2. Behavioral Measures

Participants’ performance in the MATB-II was recorded. The main variables of interest are (1) the reaction time (RT) to the events in the primary subtask (SysMon); (2) the coefficient of variation in reaction time (CV_rt_); and (3) the number of omissions made in the primary task.

#### 2.4.3. Physiological Measures

Three groups of physiological measures were recorded: EEG, eye-tracking, and electrocardiographic (ECG). However, this paper focuses only on the EEG-based physiological measures for the sake of conciseness.

The frequency-based measures extracted from the EEG signal are as follows:(1)Engagement Index (EI), as defined originally by [5], is calculated as a ratio between the power of faster brain waves, represented by the beta (β) frequency range, divided by the sum of the powers of slower brain waves, represented by the alpha (α) and theta (θ) frequency ranges, as shown in Table 1. Even though the beta band is commonly defined within the 13–30 Hz range, in our study, we have limited it to the lower beta range (13–22 Hz), following [5], and since lower beta is more typically associated with attention [55].(2)Beta over Alpha Index (shorthand: Β/A) is defined as the ratio between beta and alpha powers (β/α) (also denoted as the engagement index in some studies, such as [23]). For the purpose of comparison, as well as with the motivation of exploring measures that are feasible for practical, real-world applications, where minimizing the number of electrodes is essential to reducing setup complexity, we have computed this index over the same EEG channels as the original EI.(3)Mental Workload Index (MWI), sometimes referred to as the task load index (TLI) (e.g., [28]), is traditionally calculated as the ratio between frontal theta (Fz channel; 4–7 Hz) and parietal alpha power (Pz channel; 8–12 Hz) [13].

### 2.5. Statistical Analyses and Machine Learning Methods

The data analytics comprised a preprocessing phase of data cleaning and two kinds of analytics: analysis of variance (ANOVA) and regression. ANOVA was meant to ascertain whether the attentional demands (conditions) and their time sequence had a significant impact on the subjective-, behavioral-, and EEG-based measures. Regression was used to understand how the combination of these different groups of measures could help predict and explain the occurrence of safety-critical relevant outcome–attentional lapses, defined as the errors of omission. Since the importance of a variable in prediction depends on the model adopted, we experimented with a considerable number of library models, as detailed below.

Initial inspection of the data revealed several cases of a very large number of omissions in the primary task, indicating the neglect of the task. This disqualified 3 participants, since their number of omissions ranged from 47 to 132 SysMon events (16–45%), due to which they were excluded from all the analyses. Therefore, all the statistical analyses were conducted on 28 participants from group 1 (time-increasing demand) and 29 participants from group 2 (time-decreasing demand).

A series of mixed-measures analysis of variance (ANOVA) were performed to test the impact of attentional demands (repeated factor) and sequence of demands (between-subject factor) on EEG indices, task performance, and subjective measures, with Bonferroni correction applied for multiple comparisons. Different correlation coefficients (Pearson’s r, Spearman’s rho, and Kendall’s tau) were computed to examine the relationship between the EEG indices and behavioral (performance) measures.

Regression analysis was carried out with different methods, using a wide library of models provided by the python (v.3.10.12) library *pycaret* (v.3.3.2), which implements an AutoML approach. Indeed, the performance of a model depends crucially on the value of its hyperparameters, and the hyperparameter space is huge; this makes the tuning of a model impossible to perform simply via trial and error and requires an automatic approach: *pycaret* makes such an automatic approach available through a simple API. The regression models made available by the library are the following: extra trees regressor, Huber regressor, Bayesian ridge, ridge regression, linear regression, least angle regression, random forest regressor, light gradient boosting machine, gradient boosting regressor, K neighbors regressor, orthogonal matching pursuit, extreme gradient boosting, AdaBoost regressor, lasso regression, elastic net, lasso least angle regression, dummy regressor, decision tree regressor, and passive aggressive regressor. All those were run on our data and tuned via a random grid search (typically more efficient than an exhaustive grid search, which evaluates all possible combinations) then ranked based on typical performance metrics, such as root mean square error (RMSE) and R^2^ (coefficient of determination). The metrics used for regression performance evaluation were as follows: mean absolute error (MAE), mean squared error (MSE), root mean squared error (RMSE), coefficient of determination (R^2^), root mean squared logarithmic error (RMSLE), and mean absolute percentage error (MAPE). Their definitions are reported in Table 2.

We also investigated the importance of the input variables in predicting the target variable. To this end, we adopted a game-theory-based methodology relying on the Shapley value of the input features [56,57,58,59], which does not evaluate the predictive power of each feature in isolation but takes into account the features’ interactivity by considering all the different feature coalitions and taking the average of each feature added values. To this end, we used the SHAP (SHapley Additive exPlanations, v.0.46.0) python library implementation of the algorithm (https://shap.readthedocs.io/, accessed on 23 August 2024).

Estimating the count of omissions is a problem that can be modeled using a generalized Poisson linear regression model [60]. This type of model is typically used for modeling count data, where the outcome variable is non-negative (e.g., number of events, occurrences, etc.); in this case, occurrences of omissions in the primary task. In general, Poisson regression assumes that the mean and the variance of the target variable are equal, which is known as the equidispersion assumption. However, since the variance of omissions as the target variable is significantly higher than the mean (a condition known as overdispersion), using Poisson regression may not be appropriate because it can lead to poor model fit and underestimation of the standard errors. To avoid this issue, the model adopted is a quasi-Poisson regression, which adjusts the standard errors to account for overdispersion. It keeps the Poisson structure but relaxes the equidispersion assumption by allowing the variance to be a linear function of the mean. Since Poisson regression was not made available within *pycaret* (v.3.3.2), Poisson regression was run on the data independently using R libraries (v.4.4.1).

## 3. Results

### 3.1. Experimental Paradigm Assessment

#### 3.1.1. Experimental Manipulation: The Effect of Attentional Demands on Performance

The reaction times to the primary task increased with the increase in attentional demands, as shown in Figure 4. Analysis of variance (F (2.72, 149.55) = 173.98, *p* < 0.001; η^2^ = 0.76) showed statistically significant differences between all the conditions, except *A1 and A2* (the two baseline conditions), and *B and C* (the two dual-tasking conditions). The task load sequence and its interaction with the attentional demands did not show statistically significant effects (sequence: (F (1, 55) = 0.25, *p* = 0.62); interaction: F (2.72, 149.55) = 1.11, *p* = 0.34).

The coefficient of variation in reaction time (CV_rt_) was impacted both by attentional demands (F (2.67, 146.70) = 142.11, *p* < 0.001; η^2^ = 0.72) and the task load sequence (F (1, 55) = 19.81, *p* < 0.001; η^2^ = 0.26), while their interaction was not significant (F (2.67, 146.70) = 2.05, *p* = 0.12). In the increasing task load sequence (group 1), the differences in CV_rt_ between all the attentional demands were significant, while in the decreasing task load sequence (group 2), the only non-significant difference was that between the two baseline conditions (A1 and A2) (Figure 5). Regardless of the task load sequence, the highest CV_rt_ was observed in condition B (SysMon + Comm subtasks), followed by D (all three subtasks), and then C (SysMon + ResMan).

The average number of omissions in the primary task was relatively low, as shown in Table 3. The proportion of omissions was the highest in conditions B and D (Figure 6). Due to the low variability in the data, Box’s test of equality of covariance matrices could not be computed; therefore, the ANOVA results for the omissions are not reported here.

#### 3.1.2. Fatigue Control

As evident from Figure 4, the RTs in A2 were not significantly different from those in A1. In the increasing load sequence (G1), CV_rt_ increased in A2 compared to A1 (Figure 5, left graph), while no significant change was found for this measure in the decreasing load sequence (G2, right graph in Figure 5).

In terms of the subjectively experienced fatigue, a gradual increase was noted (Figure 7, right panel), peaking during and after the repeated baseline condition (A2). However, its values remained relatively low overall (highest average rating below 6 out of 10). A significant interaction of sequence of demands with the attentional demand conditions on fatigue ratings was found (F (3.80, 205.69) = 2.69, *p* = 0.04l η^2^ = 0.05) as a consequence of higher fatigue in group 1 during the highest attentional demand condition (D) (Figure 7, left panel).

#### 3.1.3. Mental Workload and Reported Task Difficulty

The mental workload index (MWI) reflected only a significant increase in the most demanding condition (D) compared to the least demanding ones (A1 and A2), as shown in Figure 8 (F = 5.419, *p* = 0.014; partial η^2^ = 0.090). The sequence of the demands (increasing vs. decreasing) did not significantly impact MWI (F = 2.306, *p* = 0.135), nor did it interact with the level of attentional demands (F = 1.097, *p* = 0.319).

Subjective reports of task difficulty increased with the increase in attentional demands, differing significantly between all conditions (F =122.635, *p* < 0.001; partial η^2^ = 0.679). Participants’ perceived task difficulty matched the actual levels of attentional load, except that the monotasking condition A2, which was the same as the initial baseline A1, was experienced as more difficult. The subjective level of difficulty was not affected by the sequence of the demands, and there was no interaction between them (F = 0.282, *p* = 0.597; F = 1.884, *p* = 0.127). The average ratings of task difficulty did not exceed 7/10 even in the most demanding condition.

### 3.2. Sensitivity of Engagement Indices and Subjective Reports of Attention Level

The engagement index (EI) was significantly impacted by the changes in attentional demands (F = 4.968, *p* = 0.004; partial η^2^ = 0.083) but only to a limited extent. Significant differences were found only between conditions A1 and A2, as well as between C and A2 (Figure 9, left panel). No effect of interaction nor the sequence of demands was found (F = 0.136, *p* = 0.915; F = 0.164, *p* = 0.687, respectively).

The beta/alpha ratio revealed more significant differences between the varying attention demands compared to EI, as shown in Figure 9. Importantly, successive significant differences were found between the first four conditions. Similar to EI, while the effect of the demands on beta/apha was significant (F = 6.248, *p* = 0.002; partial η^2^ = 0.102), no sequence effect was found (F = 1.050, *p* = 0.310) nor its interaction with the level of demand (F = 1.119, *p* = 0.334).

Subjectively experienced level of attention also changed significantly in response to the varying attentional demands (F = 23.245, *p* < 0.001; partial η^2^ = 0.290) but not to their sequence nor interaction (F = 0.434, *p* = 0.513; F = 0.509, *p* = 0.746; respectively). The self-reported attention level was significantly higher in the first four conditions compared to the last condition (repeated baseline), as well as compared to that after the test. After slightly longer than 20 min spent on the task, irrespective of whether the participants were exposed to a gradual (group 1) or steep (group 2) increase in attentional demands, the subjectively experienced level of attention marked a decline. This is presented visually as a transition between the lighter and darker shaded areas in Figure 10.

A summary comparison between the sensitivity of EEG indices, subjective, and performance measures is presented in Table 4. The rows correspond to *(1)* the effects of varying demands, *(2)* the direction of their change, and *(3)* the effect of fatigue onset.

### 3.3. Correlations between EEG Indices and Reaction Time Measures

Since the above-presented ANOVA results indicate that the sequence of demands (group factor) was not a significant in explaining the variability of the tested EEG indices across different experimental conditions, nor that of reaction times, the correlation analysis between those measures was conducted on the entire sample (N = 57). Positive correlations were found between the average reaction time and both the EI and beta/alpha, but not with the mental workload Index, as shown in Table 5.

A more nuanced insight into the relationship between reaction time and these indices is offered by looking into the correlation patterns within each condition (Table 6). Both the EI and beta/alpha were positively correlated with reaction times; however, significant correlation was not found within all the conditions. The increase in RTs was associated with an increase in EI within the first four conditions but not in the repeated baseline, as well as with an increase in beta/alpha, but only in conditions C and D. MWI did not correlate significantly with RTs in any of the attentional demand conditions.

Unlike all the other tested measures, CV_rt_ was significantly impacted by the sequence of demands, and its correlation with EEG measures was hence examined separately for each group. However, CV_rt_ did not correlate with any of the EEG indices in most of the attentional demand conditions. The only exception was the negative correlation between CV_rt_ and beta/alpha within condition B in the decreasing-demands group (r = −0.38, *p* < 0.05). Despite the absence of significant average correlations, the average trend of CV_rt_ corresponded with that of beta/alpha index, while this was not the case with the EI (Figure 11).

### 3.4. Triangulation of Reaction Time-, EEG-, and Subjective- Measures for Predicting the Lapses in Attention

Finally, the combined contribution of different metrics to predicting attentional lapses was examined. As mentioned earlier, the number of omissions in SysMon was taken as an indicator of lapses in attention. The regression analyses were not intended as a predictive model per se but as a method with which to understand how subjective indicators and EEG-related metrics can provide a comprehensive overview of factors influencing performance in attention-related tasks.

The correlation heat map was generated to inspect the intercorrelations among all the variables (Figure 12). Not surprisingly, the average beta/alpha and the average engagement index were highly correlated (r = 0.86). Therefore, to avoid multicollinearity issues in a generalized linear regression model, it is advisable to exclude one of these variables when constructing the model. In the presented results, beta/alpha ratio was retained as it showed slightly better results.

When considering prediction of omission in different conditions, it is important to note that conditions A1 and A2 (both containing only a single, primary task) were assigned a value of 1 to numerically indicate the level of attentional demand. Similarly, condition B was assigned a value of 2, C a value of 3, and condition D a value of 4. Performing the quasi-Poisson regression on the omissions resulted in the following Equation (1), with its corresponding coefficients and statistical significance presented in Table 7. The model obtained the equivalent of an R² = 0.419, RMSE = 1.707, MAE = 0.844.

Equation (1): quasi-Poisson regression equation for modeling the number of omissions.
(1)$$Ln\left(pMiss\right)=-7.237+0.16\;a+1.43\;b+0.05\;c+0.247\;d-0.086\;e+0.034\;f+0.145\;g.$$

Based on the model and the parameters shown in Table 7, the most predictive variable was average reaction time (avg_RT) (z = 11.44, *p* < 2 × 10^−16^), followed by the average coefficient of variation for the reaction time (avg_CV_RT) (z = 8.04, *p* = 2.59 × 10^−14^), and then the subjective difficulty (SubDiff) (z = 2.973, *p* = 0.0032), while the other predictors did not reach statistical significance.

To further explore whether other models might integrate EEG indices as meaningful predictors while avoiding the multicollinearity issue, we employed AutoML techniques, thus providing a more comprehensive comparison within the same framework. AutoML, as implemented by *pycaret*, examined the cross-validated performance of a number of regression models by searching the hyperparameter space and optimizing the parameters of the model for each set of hyperparameters. The best-performing models were then ranked according to the coefficient of determination R^2^ (representing the proportion of the variance in the dependent variable that is predictable from the independent variables). The extra tree regressor (etr) turned out to be the best performing, not only in terms of R^2^ but also in terms of the average distances MSE, RMSE, and RMSLE (metrics definitions in Table 2); the Pearson correlation coefficient between actual and predicted output value was ρ = 0.9568. Not far from the etr model were ranked a set of typically well-performing regressors: Huber’s, Bayesian, ridge, linear least angle, and random forest. The results are shown in Table 8.

We also investigated the relative importance of the input features in determining the prediction of the model, using the approximation of the Shapley value of the features provided by the SHAP python library. For the best model, the most influential feature was the average reaction time (*avgRT*), followed by its coefficient in variation (*avg_CV_RT*), then by experimental condition (*condition*). The results are shown in Figure 13, where features are ranked by mean absolute value of the SHAP values. As shown in the graph, the comparatively higher values of average RT and average CV_rt_ contribute positively to the accuracy of the prediction (helping predict higher values of omissions); analogous considerations hold for subjective ratings of task difficulty (*subDiff*) and average beta/alpha (*avg_Beta/Alpha*); on the contrary, for condition, the lower values are those that improve the accuracy.

## 4. Discussion

### 4.1. Experimental Paradigm Assessment

The proposed task-embedded reaction time paradigm involved the MATB-II task adaptation to facilitate the study of attention-level changes in a naturalistic setting while still maintaining experimental control and manipulation of variables. The results demonstrated the effectiveness of the experimental manipulation: reaction times indeed increased with the rising attentional demands. While no significant differences were observed between conditions B and C in terms of reaction times, significant differences were found between them in the coefficient of variation in reaction time; namely, a significantly higher variation in reaction times was found in condition B compared to condition C, which was further supported by a higher number of omissions in the former, indicating that these two conditions indeed pose different attentional demands.

Fatigue was successfully controlled, confirmed both by objective and subjective metrics. With respect to reaction times, no significant difference was found between the baseline condition (A1), containing only the reaction-time task, and the exact same condition repeated at the end of the task (A2), indicating a successful control of fatigue on attention and related performance. As a more sensitive measure of attentional fluctuations, the coefficient of reaction time revealed a decrease in the average performance consistency of participants that underwent the increasing demands condition (group 1). Arguably, this could indicate the very onset of fatigue effects or transient fatigue due to the high attentional demands posed by preceding condition D. However, this effect did not contribute to a significant decrease in performance, judging by the reaction time results, nor to an increase in attention lapses, judging by the comparable number of omissions in the two baseline conditions. Subjective experience of fatigue gradually increased during the experiment, showing greater sensitivity to the time on task than to the variations in attentional demands. This is evident from the only significant interaction effect observed between the demands and their sequence: fatigue ratings during the most demanding condition (D) were higher in the increasing demands group (G1) compared to the decreasing one (G2). This was due to the order of conditions; condition D was presented later in G1 compared to G2.

Finally, the task difficulty and mental workload were managed effectively. Subjective ratings of task difficulty almost perfectly reflected the experimentally designed levels of attentional demands. Interestingly, the repeated baseline condition at the end (A2) was experienced as more difficult than the first one (A2). This might also be taken as a sign of slight fatigue onset. Importantly, it indicates the susceptibility of subjective judgement to current mental state, underscoring the importance of not relying solely on subjective indicators in cognitive state assessment. Nevertheless, the average task difficulty ratings did not exceed 7/10, even in the most demanding condition, confirming that the task appeared manageable to the participants. The mental workload index proved sensitive only to the most extreme differences between the least and most demanding conditions, aligning with the task’s design to manipulate attention levels while controlling for fatigue and workload.

### 4.2. Sensitivity of Engagement Indices and Subjective Reports of Attention Level

The two EEG-based variations of the engagement index, namely, the beta/alpha and the originally proposed engagement index (EI), including theta power in the denominator, demonstrated differential sensitivity to the changes in attention levels. As shown in Table 4, which summarizes the comparative sensitivity of relevant EEG indices, performance, and subjective measures, the beta/alpha results supported higher sensitivity to dynamic variations in attention demands. Compared to EI, as well as to MWI, it captured the highest number of significant differences, explaining a greater proportion of variance in the results compared to the other two.

The EI displayed a very limited sensitivity to the variations in attentional demands. Surprisingly, it significantly increased in the repeated baseline condition (A2) compared to the first (A1) and was unexpectedly high relative to other conditions. Since A2 was the last condition in the experiment for all participants and was rather monotonous and repetitive, containing only the reaction-time task, this seemed unusual. However, this increase could indicate the effort to sustain attention and performance. A similar unexpected pattern of results regarding EI were found in the study by Kamzanova et al. [28], who also found the EI to increase towards the end of the vigilance task. Both findings seem to contradict the assumption that EI “may be more a measure of generalised alertness than of what is usually meant by engagement–a state of active task orientation” [61].

An interesting and important observation is that the significant differences between different attention demands found in the EI are complementary to those found in the beta/alpha ratio. This appears to indicate that the two indices, although similar, capture different phenomena, suggesting that they should not be used interchangeably, but rather in combination, to provide a more nuanced understanding of attention in the dynamic multitasking environments.

No effects of demands sequence on the two indices were found. This finding holds relevance for practical applications: whether there is a more demanding multitasking condition preceding a more monotonous task, or vice versa, does not affect the response of the indices. They are more sensitive to the change in the load, irrespective of the direction of change (increasing or decreasing).

### 4.3. Correlations between EEG Indices and Reaction Time Measures

The hypothesis regarding the relationship between the mental workload index and reaction times was confirmed; that is, no significant correlations were found, as the task was designed to be more sensitive to the fluctuations in attention rather than that of mental workload. However, the correlations between the EEG indices and reaction time measures displayed a somewhat unexpected pattern: both the EI and the beta/alpha correlated positively with reaction times. While the initial expectation, in accordance with the literature [23,29], was that the increase in EI would lead to faster RTs, the opposite was found. Nevertheless, this could be due to the fact that EI captures the overall level of engagement with the task at a particular moment, while the RTs indicate the level of attention specifically paid to the primary task. Therefore, the more engaged participants were with the other tasks, the longer it took them to react to the events in the primary task, yielding a positive correlation. This is supported by the finding of more significant correlations of EI and RT in the multitasking conditions compared to the single-tasking baseline conditions (A1 and A2).

Beta/alpha correlated with RT only in conditions C and D. As a reminder, both of these contained a resource managing (ResMan) task, which required constant vigilance from participants, requiring top-down attentional control, unlike the communications (Comm) task, which was activated sporadically and drew the participants’ attention away from the primary task in more of a bottom-up fashion. In the light of this observation, it seems reasonable to expect that the increase in beta over alpha power required to deal with ResMan task would be associated with the increase in reaction times to the primary task.

While no correlations were found between the EEG indices and coefficient of variation in reaction time, a similar trend in CV_rt_ and beta/alpha was found, as shown in Figure 11. However, it is important to note that these correlations were computed on the average values across each condition (5.5 mins approx.) and averaged across the entire sample or across the two groups of participants. Further examination is needed to explore how these measures relate in individual cases.

### 4.4. Triangulation of Reaction Time-, EEG-, and Subjective- Measures for Predicting the Lapses in Attention

Finally, the overall combination of metrics’ contribution to the prediction of attentional lapses was examined to test their importance in predicting real-world-related outcomes. Regression analysis was not primarily used as a prediction model but to explore how subjective indicators and EEG-related metrics might be combined to provide a more holistic view of attention-related performance in dynamic multitasking scenarios.

The results from the quasi-Poisson regression model revealed important insights into the factors influencing attentional lapses, as measured by omissions in this experiment. The model demonstrates that average reaction time (RT) and the average coefficient of variation of reaction time (CV_rt_) are the most significant predictors of the frequency of attentional lapses. Both variables show strong statistical significance, with RT having the highest predictive value, indicating that slower reaction times are strongly associated with increased omissions. Similarly, a higher variability in reaction times, as indicated by CV_rt_, also contributes significantly to predicting attentional lapses, underscoring the role of consistency in task performance as an important indicator of attention. While the instantaneous subjective ratings of attention and fatigue did not reach significance in this model, the subjective difficulty did show a statistically significant relationship with the frequency of omissions, albeit, admittedly, somewhat less pronounced compared to the reaction time metrics. Due to multicollinearity issues, the two EEG indices had to be examined separately within the quasi-Poisson model, and neither showed significant contributions to the prediction. However, it is important to note that in the present study, only the average values of these indices over the period of 5.5 min were considered, equating to the length of the entire condition. Undoubtedly, this underexploits the richness of EEG data and the variations of the indices occurring within those periods. Furthermore, in the present study, the variance of omissions was relatively low, and only the prediction of their frequency was considered. Future studies should focus on investigating the potential of these indices in predicting the omissions in real-time as a next important step towards exploring the utility of EEG-based attention related indices in safety-critical scenarios and demonstrate how transferrable these findings are to other multitasking environments.

As a further exploration of the triangulation of all the measures in understanding the occurrence of lapses in attention, AutoML was conducted, identifying the extra tree regressor (ETR) as the best-performing model, with a high Pearson correlation coefficient (ρ = 0.9568) between actual and predicted values, indicating strong predictive power for attention lapses. Similarly to the quasi-Poisson model, among the key features, average reaction time was strongly correlated with attention lapses, suggesting that slower reaction times predict a higher occurrence of lapses. Also, the coefficient of variation in reaction time was a significant contributor, emphasizing the importance of consistency in performance. Additionally, the experimental condition also played a crucial role, with lower condition values (e.g., A1 and A2, B) leading to better prediction accuracy. However, in conditions A1 and A2, where omissions were nearly absent, the model likely learned to predict this absence of lapses, a pattern not observed in condition B, which had a higher omission rate. This suggests that information about attentional demands can enhance the prediction of lapses, an important consideration for safety-critical contexts such as driving. For example, monitoring the number of driver inputs per minute could serve as an indicator of attentional demands, complementing physiological measures.

The ETR, as the best-performing AutoML model, revealed that the average beta/alpha ratio was positively correlated with predicting attention lapses, but not in the expected way. Typically associated with greater focus, the higher beta/alpha values in this study might suggest engagement with secondary tasks, which acted as distractors from the primary task, increasing the likelihood of lapses. This finding, coupled with beta/alpha ratio’s sensitivity to changes in performance consistency, indicates that it might be more suitable than the engagement index (EI) for predicting moment-to-moment fluctuations in attention, which calls for further testing.

Subjective ratings of task difficulty were also found to be associated with increased lapses, underscoring the importance of including subjective measures in predictive models. Similar to the quasi-Poisson model, the task difficulty was the only significant feature for model’s prediction among the subjective ratings. This might be due to lesser impact of biases when judging an external quality such as task difficulty, compared to self-reflections, like in estimating one’s own level of attention and fatigue.

Overall, the combination of performance metrics (RT and CV_rt_), EEG measures (beta/alpha ratio), and subjective assessments (task difficulty) provided a useful framework for understanding attention dynamics in multitasking scenarios, supporting the value of multi-modal approaches. Incorporating information about attentional demands alongside these measures further improves the prediction of the frequency of attentional lapses, emphasizing the need for a comprehensive, integrated approach in understanding attention-related tasks.

### 4.5. Study Limitations and Future Work

There are several important limitations to this study that need to be acknowledged, each of which suggests potential avenues for future research. First, while the experimental paradigm (the adaptation of the NASA MATB-II) served as a necessary step to test the task-embedded reaction time as a continuous, real-time proxy of attention level, this paradigm has not yet been validated in real-world environments. Future work should focus on replicating this framework in diverse operational settings. For example, testing this paradigm in a train-driving simulator, and eventually in real train cabins, could assess its real-world applicability for tracking train drivers’ attention in real-time; that is, embedding the measurement of train driver’s reaction time to frequently occurring vigilance alarms into the existing on-train data recorders (OTDRs) could potentially provide valuable insights into dynamic attentional fluctuations and cognitive load during driving.

A second limitation lies in the scope of the physiological measures presented in this study. Although eye-tracking (ET) and ECG data were collected alongside EEG, they were not included in the analysis due to space constraints and will therefore be presented in future papers. The decision to prioritize EEG measures was driven by their practical applicability, particularly in situations where eye-trackers may not be feasible due to privacy concerns or computational constraints. Additionally, while heart rate variability (HRV) from ECG is valuable, it reacts more slowly to cognitive changes compared to EEG, making it less suitable for real-time monitoring of attention lapses. It is essential to note, however, that these different metrics—EEG, ET, and ECG, along with subjective and performance measures—each provide unique insights into the attentional state of an operator and, ideally, should be combined in real-world applications to gain a more comprehensive understanding of attention dynamics. As emphasized by Fairclough [62], the relationship between cognitive concepts and physiological measures is complex, and rarely is there a straightforward one-to-one correspondence between them. Therefore, multimodal approaches, integrating data from various physiological sensors together with behavioral/performance and subjective data, are critical for capturing the full spectrum of cognitive processes.

In terms of EEG metrics, this study focused on commonly used indices, such as the beta/alpha ratio and the engagement index, which are prevalent in real-world applications and commercial devices for focus improvement, comparing them with the mental workload index. However, other EEG metrics, such as the theta/beta ratio or other more complex features like entropy or phase synchrony, were not explored. This limits the comprehensiveness of the current findings, as additional metrics might offer further insights into cognitive states. Future research should expand the scope to investigate these metrics.

Another limitation is related to EEG data pre-processing. While current pre-processing techniques allowed for accurate analysis, they may not be fully suited for real-time applications due to the need for rapid processing. Although the advancements in EEG technologies and computational algorithms, such as artifact subspace reconstruction (ASR), are mitigating some of the constraints of EEG data quality in real-world environments, further work should focus on streamlining pre-processing. Reducing pre-processing time would allow for more seamless and efficient use of EEG in real-time applications, especially in dynamic, real-world settings.

Additionally, the regression analyses in this study were not intended as predictive models but to explore relationships between subjective indicators, EEG metrics, and performance in attention-related tasks. This limits the generalizability of the findings. Future studies should aim to develop predictive models by utilizing shorter EEG data windows to monitor and predict performance lapses, such as omissions, in real time. These predictions could enhance safety and efficiency in operational settings, further extending the practical applications of this research.

Finally, while this study focused on detecting attention fluctuations prior to the onset of fatigue, with the aim of early detection of performance decline, future research could explore how the framework applies to tasks with prolonged attention demands where mental fatigue becomes a significant factor. Adapting this methodology to longer-duration, real-world tasks could provide further insights into attention dynamics under sustained cognitive load.

## 5. Conclusions

This study addressed the lack of methodological guidelines in neuroergonomic attention assessment for safety-critical tasks. To address this gap, we developed and validated a novel task-embedded reaction time paradigm to measure attention fluctuations in real-time. By embedding a reaction-time task into a multitasking scenario, the paradigm allowed for continuous measurement of attention, effectively bridging the gap between lab and real-world environments. The manipulation of attention levels through secondary tasks was successful, as confirmed by reaction time data.

Our hypothesis (H1) regarding the two EEG-based indices, namely, beta/alpha and engagement index (EI) was partially supported; while these indices have not increased linearly in response to the increase in demands of the task, they have nevertheless demonstrated nuanced sensitivity to the variations in attentional demands. The beta/alpha ratio was found to be effective for detecting fast, dynamic changes in attention related to arousal in high-demand scenarios, as shown by significant successive differences between the conditions. This is also supported by the similar trend of beta/alpha dynamics to those of the reaction time variability. On the other hand, the engagement index (EI) was not as sensitive to such changes but more so to mental effort in low-load, monotonous conditions, like the repeated baseline condition, where maintaining focus required increased cognitive effort. While the mental workload index (MWI) is typically considered important in conditions of cognitive overload, our results suggest the EI is valuable for monitoring attentional effort during underload conditions, such as in monotonous driving.

Both indices moderately correlated with reaction time, thus confirming the second hypothesis (H2). While some previous studies found negative correlations between these indices and reaction time, the relationship between them found in this study was positive due to the design of the experimental paradigm. EI displayed slightly higher and more consistent correlations on average compared to beta/alpha, which is in line with the conclusion that EI may reflect more the effort or executive attentional component.

With respect to the last hypothesis (H3) and the comparative usefulness of EEG and subjective measures in explaining the attention-related performance, overall, it seems that both provide different but useful information and should ideally be combined whenever possible. Combined reaction time and its coefficient of variation, the beta/alpha ratio, and subjective assessments of task difficulty proved significant in predicting the frequency of attentional lapses. Incorporating information about attentional demands alongside these measures further enhances the prediction of lapses, underscoring the need for a comprehensive, holistic approach to attention assessment in safety-critical tasks. This integrated framework provides a useful approach for understanding attention dynamics in the real-world, but further studies are needed to confirm these findings in specific safety-critical scenarios.

## Figures and Tables

**Figure 1 brainsci-14-01009-f001:**
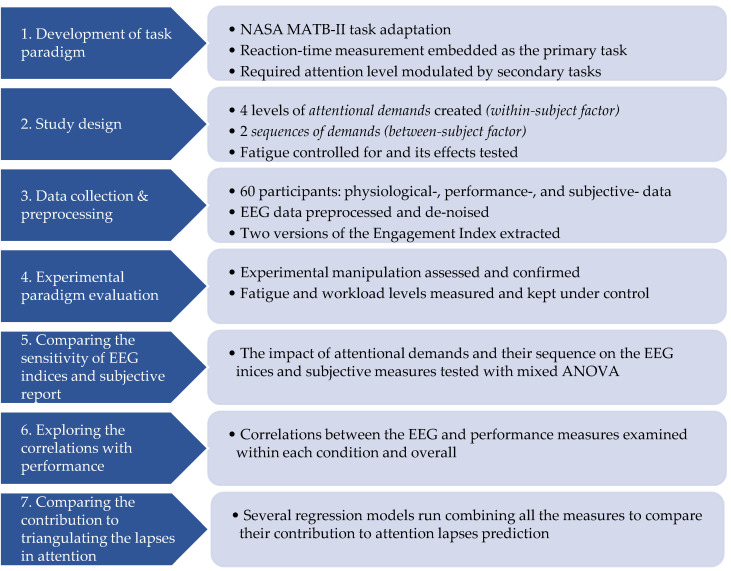
Methodological steps followed in the study.

**Figure 2 brainsci-14-01009-f002:**
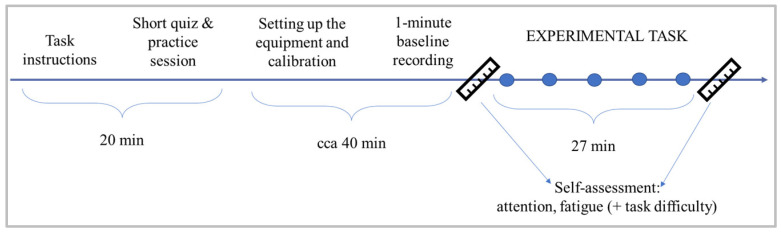
Experimental protocol and timeline.

**Figure 3 brainsci-14-01009-f003:**
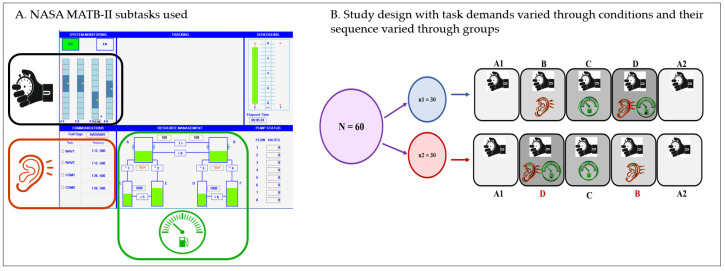
MATB-II subtasks used in this study (panel (**A**)). Task conditions representing the level of attention demanded to cope with the task (labelled A1 through A2), as well as their sequences of presentation to the two groups of participants (panel (**B**)). All conditions contain SysMon as primary task, while secondary tasks are Comm (in conditions B and D) and ResMan (in conditions C and D).

**Figure 4 brainsci-14-01009-f004:**
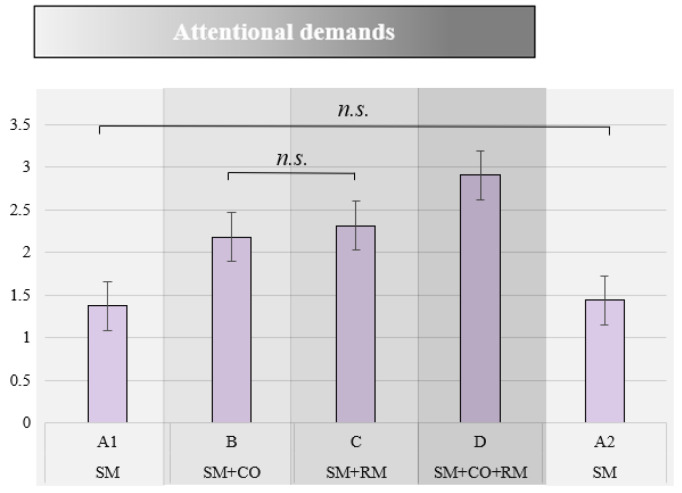
Mean reaction times in seconds (*y*-axis) in each experimental condition (n.s. denotes non-significant differences). The increase in attentional demands across conditions is represented by a grayscale, where the lightest shade corresponds to the least demanding conditions.

**Figure 5 brainsci-14-01009-f005:**
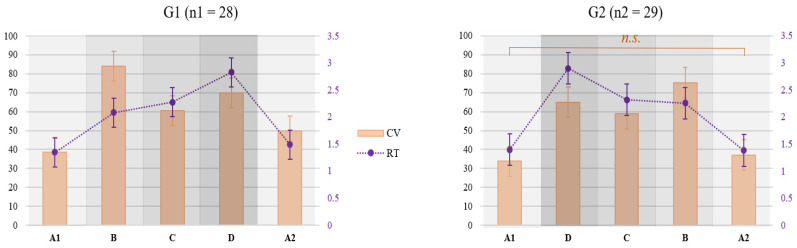
Mean coefficient of variation in reaction times across different levels of attentional demands in increasing (**left**) and decreasing (**right**) load group. Only the non-significant differences are marked. n.s. denotes non-significant differences.

**Figure 6 brainsci-14-01009-f006:**
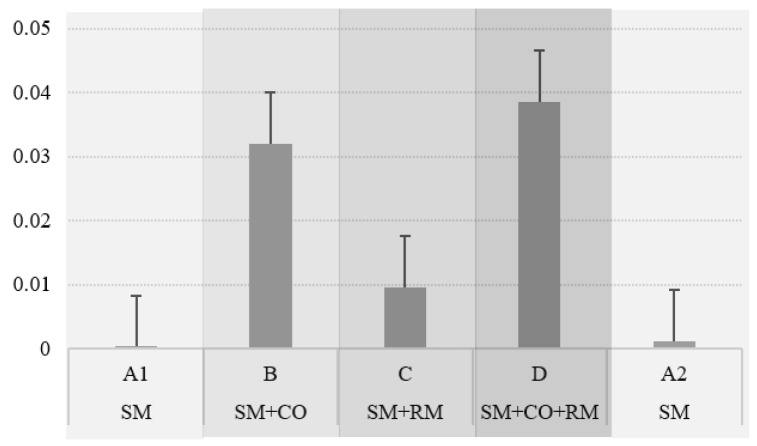
Proportion of omissions per condition.

**Figure 7 brainsci-14-01009-f007:**
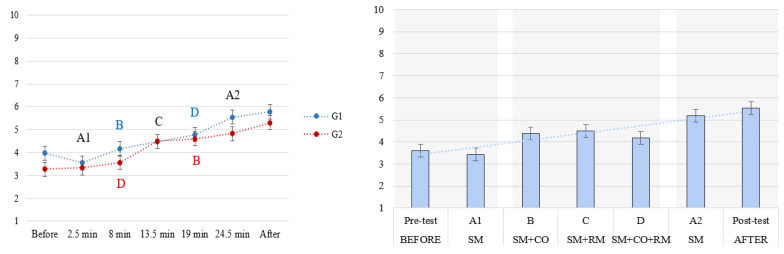
**(Right)** panel: subjective fatigue ratings; differences within each shaded block are non-significant, while those between the blocks are statistically significant. (**Left**) panel: subjective fatigue as a function of time on task.

**Figure 8 brainsci-14-01009-f008:**
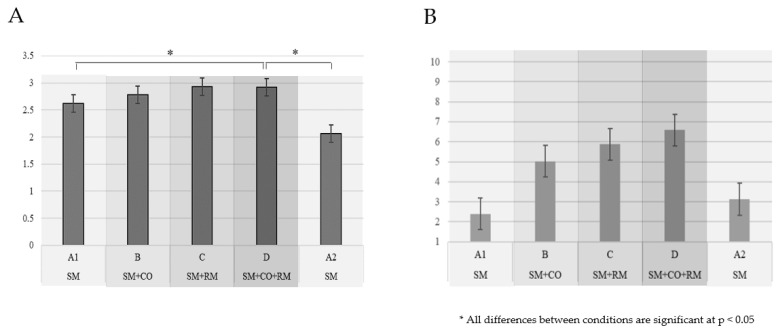
Mean mental workload index (**A**) and reported task difficulty (**B**) across the varying attentional demands.

**Figure 9 brainsci-14-01009-f009:**
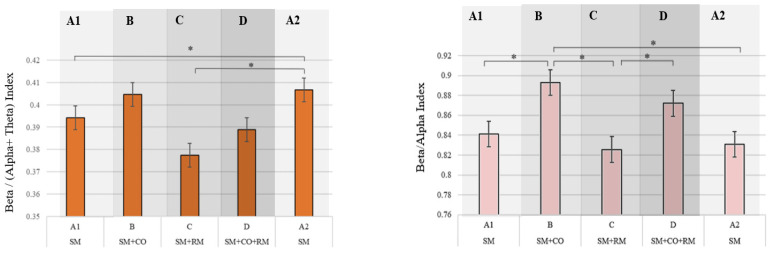
Mean engagement (**left**) and beta/alpha indices (**right**) in response to varying attentional demands. Significant differences (*p* < 0.05) are marked with an asterisk.

**Figure 10 brainsci-14-01009-f010:**
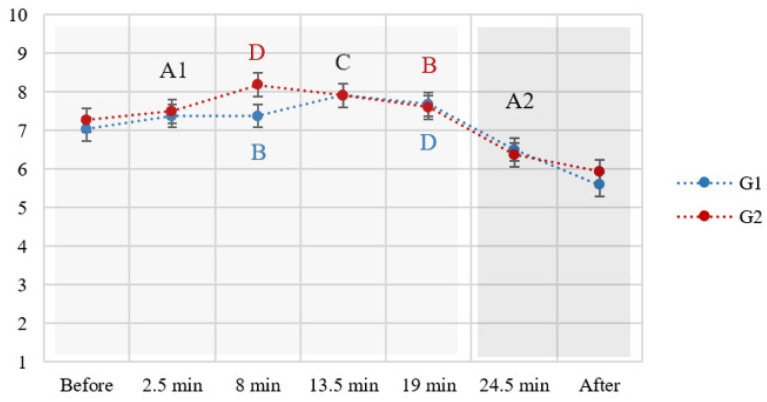
Subjective level of attention as a function of approximate time on task. The transition to darker shaded area of the graph denotes the significant decrease in ratings.

**Figure 11 brainsci-14-01009-f011:**
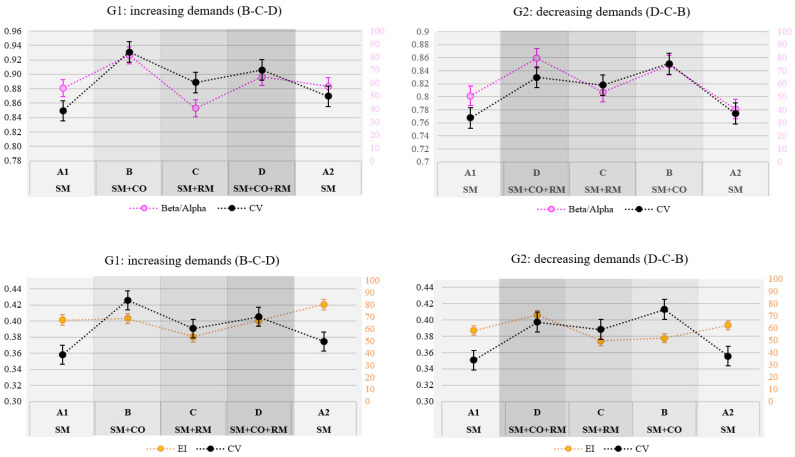
Comparison of trends in coefficient of variation in reaction time (CV_rt_) with beta/alpha (**above**) and EI (**below**).

**Figure 12 brainsci-14-01009-f012:**
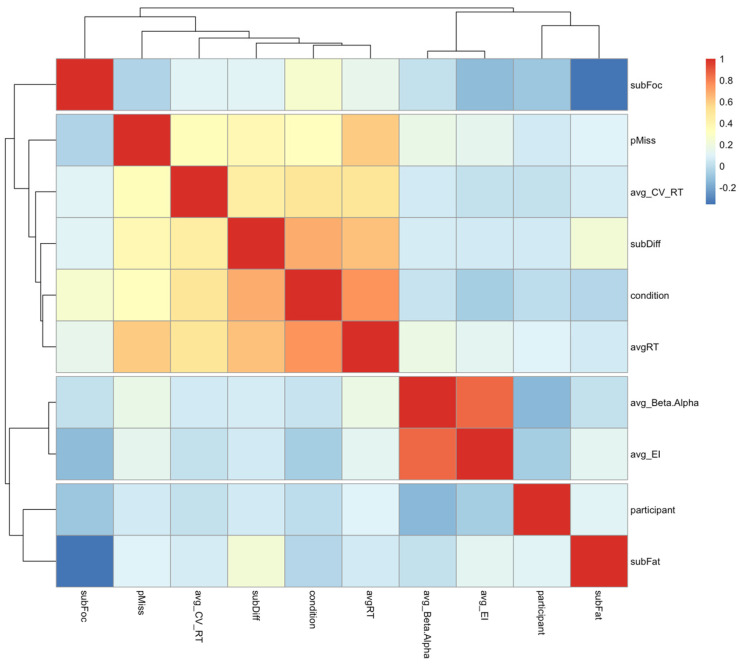
Correlation heatmap of the variables considered for the regression model.

**Figure 13 brainsci-14-01009-f013:**
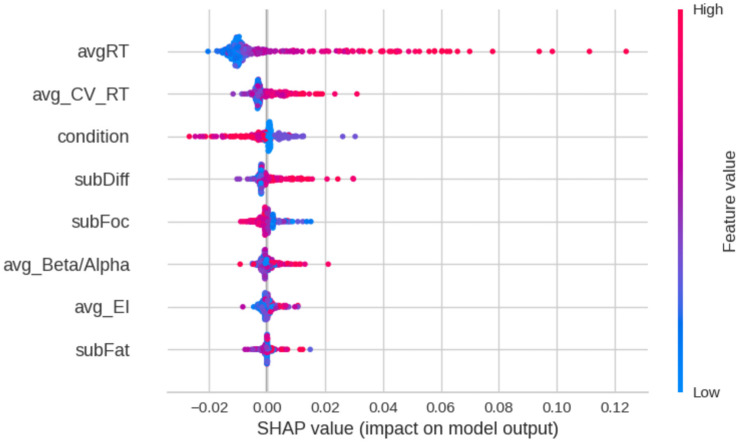
Shapley value of the input features in the task of predicting the output value through the model ert (the best of the models considered by the pycaret AutoML library). Each dot corresponds to an instance of the dataset. The color ranges from blue to red: the redder the color, the higher the value of the input feature. The impact of that feature value in improving the prediction of the model in correspondence to the specific record is represented along the *x* axis.

**Table 1 brainsci-14-01009-t001:** EEG indices calculation.

EEG Feature Name (Notion)	Formula	EEG Frequency Band Ranges (Channels)
Engagement Index (EI)	β/(α + θ)	θ: 4–8 Hz (Cz, Pz, P3, and P4) α: 8–13 Hz (Cz, Pz, P3, and P4) β: 13–22 Hz (Cz, Pz, P3, and P4)
Beta over Alpha Index (Β/A)	β/α	θ: 4–8 Hz (Cz, Pz, P3, and P4) α: 8–13 Hz (Cz, Pz, P3, and P4) β: 13–22 Hz (Cz, Pz, P3, and P4)
Mental Workload Index (MWL)	θ/α	θ: 4–7 Hz (Fz) α: 8–12 Hz (Pz)

**Table 2 brainsci-14-01009-t002:** Regression performance metrics definitions: *y_i_* denotes the true output value, ${\hat{y}}_{i}$ the predicted output value of the *i*-th point; n is the number of records.

Metrics	Definitions
Mean Absolute Error	MAE $=\frac{1}{n}\sum_{i=1}^{n}\;\left|y_{i}-{\hat{y}}_{i}\right|$
Mean Squared Error	MSE $=\frac{1}{n}\sum_{i=1}^{n}\;{\left(y_{i}-{\hat{y}}_{i}\right)}^{2}$
Root Mean Square Error	RMSE $=\sqrt{\frac{1}{n}\sum_{i=1}^{n}\;{\left(y_{i}-{\hat{y}}_{i}\right)}^{2}}$
Coefficient of Determination	$R^{2}=1-\frac{\sum_{i=1}^{n}\;{\left(y_{i}-{\hat{y}}_{i}\right)}^{2}}{\sum_{i=1}^{n}\;{\left(y_{i}-\bar{y}\right)}^{2}}$
Root Mean Squared Logarithmic Error	RMSLE $=\sqrt{\frac{1}{n}\sum_{i=1}^{n}\;{\left(\log\left({\hat{y}}_{i}+1\right)-\log\left(y_{i}+1\right)\right)}^{2}}$
Mean Absolute Percentage Error	MAPE $=\frac{100\%}{n}\sum_{i=1}^{n}\;\left|\frac{y_{i}-{\hat{y}}_{i}}{y_{i}}\right|$

**Table 3 brainsci-14-01009-t003:** The number of omissions per condition.

Condition	Min	Max	Mean	SD
A1	0.00	1.00	0.0175	0.13245
B	0.00	13.00	1.7895	2.67753
C	0.00	5.00	0.5263	1.24076
D	0.00	12.00	2.1404	2.62160
A2	0.00	2.00	0.0702	0.31958

**Table 4 brainsci-14-01009-t004:** Comparative sensitivity of EEG indices, performance, and subjective measures.

		*Reaction Time Measures*		*EEG Measures*			*Subjective Measure*
		CV_rt_	RT	β/α	EI	MWI	Self-Reported Focus Level
**Sensitivity to** **attentional** **demands**	Number of significant differences between conditions	G1: 10/10 G2: 9/10	8/10	4/10	2/10	2/10	9/21
	*Effect size (partial η^2^)*	*0.72*	*0.75*	*0.10*	*0.08*	*0.09*	*0.29*
**Sensitivity to** **direction of changes in demands**	Significant difference between increasing (G1) and decreasing loads (G2)	Yes	No	No	No	No	No
	*Effect size (partial η^2^)*	*0.27*	*-*	*-*	*-*	*-*	*-*
**Sensitivity to fatigue onset**	Significant difference between A1 and A2	Yes, in G1	No	No	Yes	No	Yes
	*Direction of difference*	*A1 < A2*	-	-	*A1 < A2*	-	*A1 > A2*

The colored columns correspond to the three groups of measures presented.

**Table 5 brainsci-14-01009-t005:** Pearson’s, Spearman’s, and Kendall’s correlation coefficients between the EEG indices and reaction time measures.

	Engagement Index (EI)	Beta/Alpha Index	Mental Workload Index (MWI)
Reaction time (RT)	r = 0.37 **	r = 0.31 *	n.s.
	rho = 0.39 **	rho = 0.34 *	n.s.
	tau-b = 0.27 **	tau-b = 0.22 *	n.s.

* and ** denote significant correlations at the 0.05 and 0.01 level (two-tailed), respectively; n.s. denotes statistically non-significant results.

**Table 6 brainsci-14-01009-t006:** Average correlations between reaction times and EEG indices within each condition.

	A1 SysMon	B SysMon + Comm	C SysMon + ResMan	D SysMon + Comm + Resman	A2 SysMon
**EI**	r = n.s.	r = 0.35 **	r = 0.32 *	r = 0.39 **	r = n.s.
	*τ* = 0.22 *	*τ* = 0.22 *	*τ* = 0.22 *	*τ* = 0.24 **	*τ* = n.s.
	ρ = 0.29 *	ρ = 0.32 *	ρ = 0.33 *	ρ = 0.37 **	ρ = n.s.
**Beta/Alpha**	r = n.s.	r = n.s.	r = 0.41 **	r = 0.32 *	r = n.s.
	*τ* = n.s.	*τ* = n.s.	*τ* = 0.23 *	*τ* = 0.22 *	*τ* = n.s.
	ρ = n.s.	ρ = n.s.	ρ = 0.36 **	ρ = 0.33 *	ρ = n.s.
**MWI**	r = n.s.	r = n.s.	r = n.s.	r = n.s.	r = n.s.
	*τ* = n.s.	*τ* = n.s.	*τ* = n.s.	*τ* = n.s.	*τ* = n.s.
	ρ = n.s.	ρ = n.s.	ρ = n.s.	ρ = n.s.	ρ = n.s.

* and ** denote significant correlations at the 0.05 and 0.01 level (two-tailed), respectively; n.s. denotes statistically non-significant results.

**Table 7 brainsci-14-01009-t007:** Coefficients and statistical significance for the quasi-Poisson regression model predicting the number of omissions.

Coefficients:	Estimate	Std. Error	z	value Pr (>|z|)
(Intercept)	−7.237976	0.846933	−8.546	8.68 × 10^−16^ ***
a = condition	−0.162389	0.096252	1.687	0.0927
b = avg_RT	1.431375	0.125078	11.44	<2 × 10^−16^ ***
c = avg_CV_RT	0.049976	0.006214	8.042	2.59 × 10^−14^ ***
d = avg_Beta/Alpha	0.247277	0.282468	0.875	0.3821
e = subFoc	−0.086203	0.051583	−1.671	0.0958
f = subFat	0.034946	0.037516	0.932	0.3524
g = subDiff	0.145577	0.048960	2.973	0.0032 **

*** and ** denote the significance levels at *p* ≤ 0.001 and *p* ≤ 0.01 respectively.

**Table 8 brainsci-14-01009-t008:** Regression models and their performance metrics. The models are ranked according to their R^2^. The best model turns out to be the extra tree regressor, also in terms of MSE, RMSE, and RMSLE.

	Model	MAE	MSE	RMSE	R^2^	RMSLE	MAPE
et	Extra Trees Regressor	0.0159	0.0008	0.0276	0.3170	0.0256	0.6836
huber	Huber Regressor	0.0163	0.0010	0.0293	0.2990	0.0272	0.5451
br	Bayesian Ridge	0.0186	0.0009	0.0284	0.2894	0.0266	0.5392
ridge	Ridge Regression	0.0187	0.0009	0.0285	0.2770	0.0267	0.5528
lr	Linear Regression	0.0188	0.0009	0.0287	0.2690	0.0267	0.5569
lar	Least Angle Regression	0.0199	0.0010	0.0298	0.1978	0.0279	0.6135
rf	Random Forest Regressor	0.0158	0.0009	0.0289	0.1903	0.0267	0.725
lightgbm	Light Gradient Boosting Machine	0.0180	0.0011	0.0313	0.1259	0.0290	0.8398
gbr	Gradient Boosting Regressor	0.0179	0.0011	0.0305	0.0925	0.0283	0.8216
knn	K Neighbors Regressor	0.0190	0.0013	0.0335	0.0700	0.0314	0.6365
omp	Orthogonal Matching Pursuit	0.0216	0.0013	0.0340	0.0458	0.0319	0.5312
xgboost	Extreme Gradient Boosting	0.0168	0.0012	0.0322	−0.0044	0.0296	0.8507
ada	AdaBoost Regressor	0.0222	0.0013	0.033	−0.068	0.0306	0.6729
lasso	Lasso Regression	0.0248	0.0015	0.0361	−0.0851	0.0339	0.5078
en	Elastic Net	0.0248	0.0015	0.0361	−0.0851	0.0339	0.5078
llar	Lasso Least Angle Regression	0.0248	0.0015	0.0361	−0.0851	0.0339	0.5078
dummy	Dummy Regressor	0.0248	0.0015	0.0361	−0.0851	0.0339	0.5078
dt	Decision Tree Regressor	0.0211	0.0018	0.0394	−0.5858	0.0363	1.0473
par	Passive Aggressive Regressor	0.0540	0.0036	0.0593	−2.9409	0.0566	1.3479

## Data Availability

The entire dataset will be publicly available upon the approval of Technological University Dublin Data Officer’s approval. It is currently in the process of being cleaned and prepared. In the meanwhile, the segment of the dataset that was used for a large portion of the analyses conducted and presented in the manuscript was attached during the submission and can be made publicly available immediately if necessary.

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
