# Peer review of "Neuroergonomic Attention Assessment in Safety-Critical Tasks: EEG Indices and Subjective Metrics Validation in a Novel Task-Embedded Reaction Time Paradigm"

_brainsci, 2024, doi:10.3390/brainsci14101009_

Round 1
Reviewer 1 Report
Comments and Suggestions for Authors
The article is devoted to the study of new methodological approaches in neuroergonomics and the role of EEG indices in assessing the stability of attention with regard to reaction time.
The article is clearly written, up-to-date and presented in a well-structured manner.
Unfortunately, throughout the text there are many places with no references to the literature (Error! Reference source not found). This requires considerable revision of the text by the authors. However, here a technical errors in the drafting of the text are possible.
The reference list is up-to-date and includes more than 50% of the sources 2019-2024 (last 5-6 years). However, the authors should note the source [22] (J. Prinsen, E. Pruss, A. Vrins, C. Ceccato, and M. Alimardani...), which does not contain the year of publication.
No excessive self-citation by the authors has been identified.
The manuscript is written in scientific language and the design of the experiment is consistent with the hypothesis being investigated.
The results can be reproduced based on the applied methods.
The figures and tables are quite revealing.
Statistical analyses are presented in great detail.
Conclusion is very widely presented, some of the material repeats the discussion or should be included in the Discussion section. The conclusion should be more brief and reflect findings on whether the hypotheses are confirmed (H1, H2 H3).
At the end of the Discussion section, it is recommended that the authors provide a subsection on the limitations of the study. For example, this research does not reflect possible gender differences in the features of attention and mental work efficiency in multitasking conditions.
Ethical principles were adequate to the design of the study.
Reviewer 2 Report
Comments and Suggestions for Authors
This article titled "Neuroergonomic Attention Assessment in Safety-Critical Tasks: EEG Indices and Subjective Metrics Validation in a Novel Task-Embedded Reaction Time Paradigm" explores the use of EEG-based measures to evaluate attentional demands in multitasking environments, particularly those critical to safety, such as driving or automated systems. The study introduces a novel reaction-time paradigm embedded within the NASA Multi-Attribute Task Battery (MATB-II), manipulating attentional demands through secondary tasks. The findings suggest that both indices are sensitive to changes in attentional demand, with Beta/Alpha ratio proving more responsive. This work highlights the potential of combining EEG metrics and subjective assessments for better real-time monitoring of attention lapses, offering insights for applications in safety-critical industries. Following are my major concerns regarding this article:
1. In Section 1.2 (Research Overview), the validation of EEG indices such as the Engagement Index (EI) and Beta/Alpha (B/A) ratio appears limited. The study does not compare these indices to other well-established metrics used in similar real-world applications, which affects the generalizability of its findings in broader contexts.
2. In Section 2.2 (Experimental Paradigm), the study attempts to control fatigue by limiting the task to under 30 minutes, but this may not be sufficient for safety-critical tasks that demand sustained attention over longer periods. The short duration limits the understanding of how the EEG indices would behave under conditions of prolonged mental fatigue, which is common in real-world tasks.
3. In Section 2.4.1 (Subjective Measures), reliance on self-reported focus and task difficulty introduces potential bias. While subjective measures are common, there are no additional objective metrics, such as eye-tracking or heart rate variability, to corroborate or cross-validate the self-reported data. This limits the reliability of subjective assessments.
4. In Section 2.4.3 (Physiological Measures), the focus on time-domain measures (e.g., Engagement Index and B/A ratio) excludes other valuable EEG analyses such as connectivity or network-based measures. These could provide deeper insights into how different brain regions coordinate under varying attention levels, enriching the understanding of neural mechanisms.
5. In Section 3.2 (Sensitivity of Engagement Indices), the regression models used to predict attention lapses (e.g., Extra Trees Regressor) are not thoroughly tested for generalization across different task types. It is unclear whether the models would maintain their predictive accuracy in diverse multitasking scenarios, and further cross-validation with different datasets is needed to ensure robustness.
Round 2
Reviewer 2 Report
Comments and Suggestions for Authors
The authors have significanlty addressed all my comments. I have no further concerns. I recommend accepting this article for publication.
Best of luck to the authors.